# Nutrients and non-essential metals in darkibor kale grown at urban and rural farms: A pilot study

Brent F. Kim[1,2], Sara N. Lupolt[1,2,3], Raychel E. Santo[1,2], Grace Bachman[1], Xudong Zhu[4], Tianbao Yang[4], Naomi K. Fukagawa[5], Matthew L. Richardson[6], Carrie Green[7], Katherine M. Phillips[8]*, Keeve E. Nachman[1,2,3,9]*

1 Johns Hopkins Center for a Livable Future, Johns Hopkins Bloomberg School of Public Health, Baltimore, Maryland, United States of America, 2 Department of Environmental Health & Engineering, Johns Hopkins Bloomberg School of Public Health, Baltimore, Maryland, United States of America, 3 Risk Sciences and Public Policy Institute, Johns Hopkins Bloomberg School of Public Health, Baltimore, Maryland, United States of America, 4 US Department of Agriculture, Food Quality Laboratory, Agricultural Research Service, Beltsville, Maryland, United States of America, 5 US Department of Agriculture, Beltsville Human Nutrition Research Center, Agricultural Research Service, Beltsville, Maryland, United States of America, 6 Center for Urban Research, Engagement and Scholarship, University of the District of Columbia, Washington, DC, United States of America, 7 US Department of Agriculture, Adaptive Cropping Systems Laboratory, Agricultural Research Service, Beltsville, Maryland, United States of America, 8 Department of Biochemistry, College of Agriculture and Life Sciences, Virginia Tech, Blacksburg, Virginia, United States of America, 9 Department of Health Policy and Management, Johns Hopkins Bloomberg School of Public Health, Baltimore, Maryland, United States of America

* knachman@jhu.edu (KEN); kmpvpi@vt.edu (KMP)

**Data Availability Statement:** All relevant data are within the paper and its Supporting information files.

## Abstract

Kale is a nutrient-dense leafy vegetable associated with wide-ranging health benefits. It is tolerant of drought and temperature fluctuations, and could thus serve an increasingly important role in providing a safe and nutritious food supply during the climate crisis, while kale's ease of cultivation and ability to be grown in a wide range of soils make it a good fit for urban agriculture. In this pilot study we explored potential differences between kale grown at urban versus rural farms. We planted kale seedlings (Darkibor variety) at three urban and four rural farms in and around Baltimore City, Maryland, instructed farmers to cultivate them using their usual growing practices, harvested the kale from fields and points of distribution, and analyzed it for concentrations of carotenoids, vitamins C and $K_1$, ten nutritional elements, and eight non-essential metals. Although sample sizes for some analyses were in some cases too small to produce statistically significant results, we identified potentially meaningful differences in concentrations of several components between urban and rural kale samples. Compared to urban samples, mean concentrations of carotenoids and vitamins were 22–38% higher in rural field samples. By contrast, mean concentrations for eight nutritional elements were higher in urban field samples by as much as 413% for iron. Compared to rural field samples, mean concentrations of nine non-essential metals were higher in urban samples, although lead and cadmium concentrations for all samples were below public health guidelines. Some urban-rural differences were more pronounced than those identified in prior research. For six elements, variance within urban and rural farms was greater than variance between urban and rural farms, suggesting urbanicity may not be the

**Funding:** This work was supported by cooperative agreement 58-8040-8-018 between the US Department of Agriculture Agricultural Research Service and Virginia Tech, and by cooperative agreement 58-8040-8-021 between the U.S. Department of Agriculture Agricultural Research Service and Johns Hopkins University. Sara Lupolt was supported by a dissertation grant from the Johns Hopkins 21st Century Cities Initiative; a pilot award from the Johns Hopkins Education and Research Center for Occupational Safety and Health, supported by the National Institute for Occupational Safety and Health and the US Department of Agriculture Northeast Sustainable Agriculture Research and Education Program (GNE 19-209); and a Johns Hopkins Center for a Livable Future-Lerner Fellowship. Because this was a cooperative agreement, three USDA staff members (Naomi Fukagawa, Carrie Green, and Tiabao Yang) were involved in helping design, conduct, and co-author the study. Other funders had no role in preparing, reviewing, or editing the manuscript. Support for statistical consulting was made possible by The Johns Hopkins Institute for Clinical and Translational Research, funded in part by [Grant Number UL1 TR003098] the National Center for Advancing Translational Sciences, a component of the National Institutes of Health (NIH) and the NIH Roadmap for Medical Research. Because this was a cooperative agreement, three USDA ARS staff members (NF, CG, and TY) were involved in helping with study design, data collection and analysis, and preparation of the manuscript. All other funders had no role in study design, data collection and analysis, decision to publish, or preparation of the manuscript.

**Competing interests:** The authors have declared that no competing interests exist.

primary driver of some observed differences. For some nutrients, mean concentrations were higher than upper ranges reported in prior estimates, suggesting kale may have the potential to be more nutrient-dense than previously estimated. The nutritive and metals composition of this important crop, and the factors that influence it, merit continued investigation given its growing popularity.

## Introduction

Kale (*Brassica oleracea* var. *acephala*), widely considered a "superfood," is a nutrient-dense leafy vegetable associated with antioxidant, anticancer, cardiovascular, and gastrointestinal benefits [1]. According to United States Department of Agriculture (USDA) FoodData Central Foundation Foods data [2], and relative to US Food and Drug Administration (FDA) recommended daily values (DV) [3], a standard serving of 65 grams of raw kale provides over 200% of the DV for vitamin $K_1$ and over 20% of the DV for vitamin C and manganese (over 20% is considered "high in" a nutrient by FDA standards), and 10–20% of the DV for vitamin A, riboflavin, calcium, and folate (10–19% is considered a "good source"). Even among cruciferous vegetables—known for their nutrient density—kale has been reported to be exceptional in its content of many vitamins and minerals [1]. Beyond essential nutrients, kale contains health-promoting phytochemicals in the glucosinolates, polyphenols, and carotenoids groups [1, 4].

The popularity of kale has surged in recent decades. *Bon Appétit* magazine named 2012 the year of kale; a year later, "National Kale Day" was established in the US [5]. Between 1997 and 2017, US kale production and supplies nearly tripled, with per capita availability peaking at 467 grams (over one pound) per US citizen annually. Production and availability have since declined slightly but remain well above historical levels [6]. Countries outside the US, such as Denmark [7] and Australia [8], have reported a similar rise in demand. Kale's growing popularity extends beyond fresh and frozen products; the global market for dehydrated kale chips, for example, is projected to more than double between 2020 and 2027 [9]. At a more local level, according to a 2016–2017 survey of 104 urban farms and community gardens in Baltimore City, Maryland, kale was grown at 34 percent of sites and was the 3rd most frequently grown produce item after tomatoes and peppers [10]. It has been suggested that kale's popularity may be buoyed in part by the plant's tolerance of unfavorable agronomic conditions, making it more popular among farmers during a rapidly changing climate [1]. Beyond uses in human food, Brassica plants have a history of use in traditional medicine [1], and older kale leaves may be used as animal fodder [11].

The nutritive value and widespread popularity of kale are grounds for investigating the many factors that affect kale's beneficial properties. For example, studies have demonstrated that the nutrient and/or phytochemical composition of kale can vary based on cultivar [1, 12–15], species of cover crop [16], agrochemical use [15], fertilizer composition and application rate [17, 18], moisture stress [19], short-term exposure to low temperatures [20], growing season, [15, 21], maturity stage (e.g., microgreens vs. adult leaves) [14, 22, 23], time between planting and harvest [24], and processing method (e.g., drying) [25]. Leaf biomass has been shown to be influenced by many of the same factors, including cultivar [24, 26], moisture stress [19], and time between planting and harvest [24, 26].

Environmental factors can also impact concentrations of non-essential metals and other harmful contaminants in kale. Urban areas, for example, often have high concentrations of industrial activity, waste incineration, building demolition, lead-based paint, vehicular

emissions, tire wear, and other sources of harmful metals [27–31]. Releases from these sources may contaminate growing soils via various environmental pathways, including runoff and airborne deposition [29, 31], and subsequently be taken up by plant tissues. Uptake rates may vary widely by compound, plant species, and the part of the plant, e.g., roots vs. leaves [32–34].

Generally, the health benefits of consuming vegetables likely outweigh any risks associated with exposure to non-essential metals and other harmful contaminants; regardless, research and monitoring are important to ensure the safety of food supplies. An assessment of produce grown in Baltimore City, Maryland found some significant differences in concentrations of essential and non-essential (arsenic (As), barium (Ba), cadmium (Cd), chromium (Cr), lead (Pb)) metals in kale samples from urban-grown sources compared to peri-urban, grocery conventional, and grocery organic sources; however, differences were too small in magnitude to have any practical significance for health, and Cd and Pb concentrations in all samples were well below public health guidelines for exposure (no guidelines are available for As, Ba, and Cr) [35]. A San Francisco, California study found no significant differences in Cd or Pb concentrations in kale from urban, suburban, or grocery sources, and all levels were below public health guidelines [36], and a North Carolina study found only 20 percent of kale samples from commercial sources had lead concentrations above their detection limits [37].

Despite its popularity, raw kale is not included in the FDA's Total Diet Study [38], which monitors levels of nutrients and contaminants in common foods (the 2018–2020 dataset for the first time added a pan-cooked version of kale).

Building upon the existing body of evidence, we conducted a pilot study to assess levels of select health-relevant nutrients, nutritional elements, and non-essential metals in kale grown at two urban and four rural farms; and to gather data about site history, farming practices, environmental conditions, and other factors that might influence the properties of kale grown at those sites. The aim of the study was to explore the degree to which concentrations of these analytes might differ between urban and rural farms, and possible reasons for those differences. The analytes of interest were carotenoids, vitamin $K_1$, and vitamin C; nutritional elements calcium (Ca), copper (Cu), iron (Fe), magnesium (Mg), manganese (Mn), molybdenum (Mo), phosphorous (P), potassium (K), sodium (Na), and zinc (Zn); and non-essential metals As, Ba, Cd, Cr, Pb, nickel (Ni), uranium (U), and vanadium (V).

## Materials and methods

### Farm selection

We identified and recruited seven farms for participation in this study. Three urban farms were identified through previous inclusion in the Safe Urban Harvests Study [10, 35]. Four rural farms, including a farm established to research and test small-scale sustainable agricultural practices, were identified via the study team's professional networks. We distinguished urban farms from rural using 2010 Maryland population density as a rough proxy for urbanicity; urban farms were in census tracts with over 4,000 people per square mile, whereas rural farms where in tracts with fewer than 1,001 [39]. Urban farms were located within Baltimore City; rural farms were within one hour of driving distance from Baltimore City. All farms had previous experience growing at least two varieties of kale, e.g., Red Russian or Winterbor. For reporting results, farms were anonymized by assigning a unique identifier beginning with "U" for urban farms and "R" for rural farms.

### Kale seedlings and transplanting

For the purposes of this survey study, one variety of kale, Darkibor, was selected. Organic (F1) Darkibor kale seeds were acquired from a commercial seed vendor (Johnny's Selected Seeds,

Maine, US). The seeds were sown in seedling starting trays filled with Fafard growing mix (Sun Gro Horticulture, MA, US) on 25 July 2019 in a growth chamber (temperature 25°C, relative humidity 60%, in darkness) at the US Department of Agriculture (USDA) Agricultural Research Service's Beltsville Agricultural Research Center laboratory in Beltsville, MD. Germination of the seeds was first noted two days after sowing. Plants were then grown under 14h/10h light/dark with an intensity of 160 μmol per $m^2$ second. After one month, seven trays of 32 seedlings each with similar growth were selected for use in the study.

Between 27–29 August 2019, we transplanted 32 seedlings at each farm using stainless steel trowels at a spacing of 12 inches (30 cm) in an area determined by the farmer, in configurations most suitable for the location, e.g., 1 row of 32 seedlings, 2x16, or 4x8. All plants were grown in open air, i.e., not in high tunnels/hoop houses. Farmers were otherwise instructed to grow and tend to the kale plants per usual practices for their farm (e.g., some farms applied pesticides, some installed and used row cover). Farmers were compensated $1,000 USD for the use of space, labor, and time in growing the kale.

## Surveys on farm history and growing practices

At the time of transplanting, members of the study team verbally administered a baseline survey (provided in Supporting Information) to a representative at each farm. The survey included questions about farm history; prior soil testing for fertility or contaminants; and growing practices, e.g., irrigation, pest management, use of soil amendments, and USDA Organic certification.

After harvesting, a follow-up survey was emailed to the representative at each farm. The survey asked representatives to verify which practices—specifically irrigation, pest management, and use of soil amendments—had been used to grow the study kale and whether those differed from their typical kale growing practices reported in the baseline survey.

## Kale harvesting and collection

After at least one farmer notified the study team that their kale was ready to harvest (approximately six to seven weeks after transplanting), four kale samples were collected directly from each farm ("field samples") between 11–18 October 2019. The 12 largest and healthiest plants at each farm, selected based on a visual assessment of size and the fewest spots and yellow leaves, were harvested, and sets of three plants were randomly selected and combined into four composite samples. Two study team members took a photo of the plants prior to harvesting, removed each plant from the ground (including roots) using a shovel, and recorded the time of harvest. Each sample was placed in a five-gallon (approximately 19-liter) bucket and promptly transported via air-conditioned vehicle to a laboratory at the Johns Hopkins Bloomberg School of Public Health (BSPH), where it was immediately processed.

During the same week, two samples of kale from each farm were collected from the point of distribution ("market samples"), i.e., farmers market, mobile market, or donation site. Farmers obtained six bunches of market-quality kale, which were then randomly assigned to two composite samples of three bunches each. The kale in the market samples was harvested, processed, and transported to the point of distribution, following the usual practices for that farm (e.g., any washing, sanitizing, use of refrigerated storage and/or delivery). The study team picked up the harvested samples at the point of distribution and transported them to the BSPH laboratory, where they were immediately processed. When study team members picked up market samples, they verbally administered a market survey (provided in Supporting Information) to a representative of each farm. The survey included questions about the date and time of harvest, processing (e.g., washing), and use of refrigeration during transport.

## Kale sample processing and storage

Upon arrival at the BSPH laboratory, each sample was inspected by the study team. Leaves deemed edible by the study team (e.g., not yellow or covered in pests) were separated from the central stem and roots. Each sample (i.e., leaves without the central stem and roots) was then weighed, rinsed in deionized water, patted dry with WypAll lint-free surface wipes, laid out on a flat white surface, photographed, and weighed again.

The center rib of each leaf was removed with a stainless-steel knife on a plastic, consumer-grade kitchen cutting board. Following previously described procedures [40] established for analysis of produce samples for the USDA National Food and Nutrient Analysis Program [41] and FoodData Central Foundation Foods datatype [42], leaves were then placed in stainless steel bowls with liquid nitrogen, flash frozen, and broken into smaller pieces. Flash-frozen samples were weighed again. The flash-frozen samples were immediately processed in a Blixer V6 industrial food processor (Robot-Coupe USA, Inc., Ridgeland, MS) for 70 seconds or until reaching the consistency of a fine powder, adding additional liquid nitrogen if necessary to keep the material frozen. The powdered homogenate from each sample was aliquoted into multiple labeled 60-mL glass jars (pre-cleaned and certified to meet EPA guidelines for environmental sampling [43]) (Environmental Express, Charleston, SC), sealed, wrapped in foil to prevent light exposure, and stored in a -80°C freezer until distributed for analysis. Between each sample, all processing equipment was washed with Alconox detergent and warm water, air-dried, and rinsed with acetone and allowed to dry.

## Nutrient and non-essential metals analyses

Moisture was analyzed in all samples to assess differences on a fresh and dry mass basis. Ten elements essential for human health were selected for analysis: calcium, copper, iron, magnesium, manganese, molybdenum, phosphorous, potassium, sodium, and zinc. Vitamin C, vitamin $K_1$, and carotenoids contents were additionally analyzed since kale is recognized as a good dietary source of these nutrients, and were nutrients for which methodology and analytical precision had been established in previous studies to be sufficient for detecting meaningful sample-to-sample variability [40, 44, 45]. Eight non-essential metals (contaminants of public health concern) were also selected for analysis: arsenic, barium, cadmium, chromium, lead, nickel, uranium, and vanadium. These contaminants may be present in plant tissues and/or on plant surfaces as a result of airborne deposition, soil uptake, or other pathways [35]. Chromium was not speciated between hexavalent chromium (a carcinogen) and trivalent chromium (an essential nutrient).

Samples were packed in dry ice after removal from the -80°C freezer at the BSPH laboratory, shipped via overnight express to designated laboratories, verified to have arrived frozen upon receipt at each laboratory, and held at -60°C storage until analyzed. Vitamin C and moisture were analyzed at Virginia Tech (Phillips laboratory), vitamin $K_1$ at Tufts (USDA Human Nutrition Research Center on Aging at Tufts University, Boston, MA), carotenoids at Eurofins-Craft Technologies (Wilson, NC), and nutritional elements and non-essential metals at Eurofins Scientific (Madison, WI).

**Analytical methods.** Moisture was measured by vacuum drying 2 g subsamples to a constant weight at 635 mm Hg and 65–70°C, as adapted from Association of Official Analytical Chemists International [46]. Vitamin C was analyzed as total ascorbic acid after reduction of dehydroascorbic acid with tris (2-carboxyethyl) phosphine hydrochloride, by reversed-phase high-performance liquid chromatography, as previously described [47], using 2.0±0.1 g analytical subsamples that were weighed while still frozen. Stability of vitamin C in homogenized raw vegetables analyzed under the conditions of this study has been validated [44, 45]. For

carotenoids (alpha-carotene, beta-carotene, gamma-carotene, alpha-cryptoxanthin, beta-cryp-toxanthin, lutein, lycopene, and zeaxanthin), samples were extracted and analyzed as previously described using high-performance liquid chromatography (HPLC) for quantitation [48]. For vitamin $K_1$, extracts were prepared using hexane and purification by silica solid phase extraction, and analyzed for phylloquinone (vitamin $K_1$), menoquinone-4, and dihydrophyllo-quinone using reversed-phase HPLC with fluorescence detection and with vitamin $K_{1(25)}$ as an internal standard [49]. Nine elements (Ca, Cu, Fe, K, Na, Mn, Mg, P, and Zn) were analyzed by inductively coupled plasma emission spectroscopy (ICP) after digestion with concentrated hydrochloric acid [50]. The remaining elements (As, Ba, Cd, Cr, Pb, Mo, Ni, U, V) were analyzed by ICP with mass spectrometry detection (ICP-MS) after digestion with concentrated nitric acid and water using a closed-vessel microwave digestion system [51, 52].

**Analytical quality control.** For each nutrient, 10–15% of the samples were assayed in duplicate with the laboratory blinded to the sample duplicates. Additionally, well-characterized in-house control materials, developed for the USDA National Nutrient Database/Food Data Central [42] ("CC") and/or commercially available certified reference materials ("RM") were included in each assay batch, as follows. For vitamin $K_1$, a mixed vegetable composite ("Vegetable III CC"), and SRM® 3232 Kelp Powder and SRM® 1869 Adult/ Infant Nutritional Formula II from National Institute of Standards and Technology (NIST) (Gaithersburg, MD; [53]); for carotenoids, Vegetable III CC and NIST SRM® 1869; for vitamin C a mixed vegetable/fruit composite ("Mixed Vegetable II CC") and BCR CRM 421 Milk Powder [54] purchased from Sigma-Aldrich (St. Louis, MO); for moisture, Mixed Vegetable II CC; for nutritional elements, Vegetable III CC and NIST SRM® 2383a Baby Food; for metals and Mo NIST SRM® 1515 Apple Leaves (Ba, Cd, Cr, Mo, Ni, Pb, U, V), NIST SRM® 1570a Spinach Leaves (As, Cd, Ni), NIST SRM® 2383a Baby Food (Ba, Cr, Ni), and Vegetable III CC. The CC had established tolerance limits for quality control purposes, as described previously [55].

## Environmental data collection

At the time of seedling transplanting and weekly thereafter until harvesting, two members of the study team visited each farm to collect data on environmental conditions, totaling six visits over five weeks. During each visit, a DustTrak portable aerosol and dust monitor (TSI, Shoreview, MN) was used to measure mean concentrations of particulate matter ($PM_1$, $PM_{2.5}$, respirable PM, $PM_{10}$, and total PM) over a five-minute period. Weekly rainfall amounts, collected by consumer-grade rain gauges, were also recorded (in cm). Photographs of kale plants were taken by the study team each week and any substantial changes or observations were noted.

On the first visit after transplanting, three Bluetooth-enabled HOBO data loggers (ONSET, Bourne, MA) were installed within each kale plot. Two were installed above ground to track light intensity and ambient air temperature. One was buried at root depth to track soil temperature. Loggers recorded light and temperature data at five- to 15-minute intervals from installation until they were stopped and retrieved at the time of harvesting.

## Data reporting and analyses

Data management, analysis, and visualization were performed using Python version 3.6. For some visualizations Microsoft Excel was additionally used.

For some non-essential metals (As, Ni, Pb, U, V), results for some samples were below the detection limit (5, 10, and 50 ppb for lead and uranium, vanadium and arsenic, and nickel, respectively; see S2 Table in S1 Data for results). Prior to the following analytical

steps, including calculating means of duplicates, any values below the detection limit were assumed to be at the detection limit. This assumption slightly overestimates concentrations of these metals; e.g., if a value for arsenic was below the detection limit, it was assumed to be 10 ppb.

For samples analyzed in duplicate for a given component, the mean was used as the sample result.

Analyte concentrations were reported by laboratories on a per fresh weight basis, e.g., mg vitamin C per 100 g fresh kale. Results herein are also reported on a per fresh weight basis, since this is the form in which kale is typically consumed, i.e., not dehydrated. In the Supporting Information and Supporting Data Tables we additionally present results after converting to dry weight, to control for moisture content. Fresh weight concentrations were converted to dry weight using the following formula adapted from the US Environmental Protection Agency (EPA) [56]:

$$C_{a,i,dry} = \frac{C_{a,i,fresh}}{1 - W_i}$$

Where $C$ is the concentration of analyte $a$ in sample $i$ and $W$ is the wet fraction for sample $i$.

For statistical analyses, sample descriptive variables included an identifier for each farm, farm type (urban, rural), and harvest location (field, market). Two-sided Mann-Whitney U tests were used to compare analyte concentrations between groups, e.g., samples from urban vs. rural farms, samples collected from farms vs. at market, or samples from farms using a particular growing practice vs. those that did not. One-way analysis of variance (ANOVA) was also used to compare differences between urban and rural samples; although this test is more appropriate for larger sample sizes and normally distributed data, it is useful for comparing variation between urban and rural samples to variation within urban and rural samples. Potential correlations between continuous variables, such as between different analyte concentrations, were assessed using Pearson's correlation tests.

Data from individual kale samples were clustered, i.e., we would expect analyte concentrations among kale samples from the same farm to be correlated, thus kale samples were not independent. To account for clustering, for the purpose of statistical testing with kale data we used the mean value for each cluster (i.e., farm), as previously recommended [57].

Analyses of light and temperature were based on hourly means and included only those data that were common to every group (e.g., when checking for differences across farms, if a HOBO data logger at one farm was started 30 hours before those at the other farms, those first 30 hours were excluded from analyses). Estimates of peak hourly light and temperature for each day excluded days that did not have a full 24 hours of data, i.e., days when data loggers were either installed or removed. Wilcoxon signed-rank tests were used to compare two groups with repeated measures over time, i.e., PM concentrations, weekly collected rainfall, and mean hourly light and temperature between urban and rural farms.

With the exception of ANOVA, the aforementioned tests are all non-parametric and were used on the rationale that sample sizes were small. Light and temperature data had larger sample sizes, e.g., there were 907 mean hourly light intensity observations per farm; however, the results of Shapiro-Wilks tests indicated that the distributions of light and temperature data were non-normal and thus better suited for non-parametric testing.

## Ethical considerations

The Johns Hopkins Bloomberg School of Public Health Institutional Review Board (IRB) reviewed and determined that this study did not require oversight as human subjects research.

## Results

### Kale yields

Thirty-two kale samples were represented in the results (Table 1). These comprised 23 field samples and nine market samples, including four field samples and two market samples from each rural farm. After harvesting four field samples from urban farm U1, there was only enough kale remaining for one market sample. Urban farm U2 had a harlequin bug (*Murgantia histrionica*) infestation and only had enough kale for three field samples and no market samples. A third urban farm encountered significant pest issues that inhibited kale production; no kale was harvested from this farm and their survey results were excluded from the study. Sample weight, measured after removing the central stem and roots, washing, and drying, ranged from 319–1840 grams (mean: 923 g). S1 Fig in S1 File shows photos of cultivated kale at each farm.

### Farm history and growing practices

Four of the six eligible farms were established between 2009–2018, while two rural farms (R3, R4) were established in the 1980s. The urban farms were previously vacant lots with some history of residential use. The rural farms all had some history of use as a farm or garden prior to current management and their participation in this study.

Both urban farms had previously tested their soil for metals as part of their participation in the Safe Urban Harvests Study [35]. One urban farm (U2) reported testing their soil for metals annually. Of the rural farms, only the research farm (R4) had tested their soil for metals and repeated testing every three to four years. One urban (U2) and three rural farms (R2, R3, R4) had tested the fertility of their soil (e.g., nutrients, pH).

All farms had at least two years of experience growing kale, and all reported growing at least two different varieties of kale since the establishment of their respective farms. Only one rural farm (R3) had prior experience growing the Darkibor variety of kale. All rural farms reported using cover crops; the urban farms did not. Other crops previously grown on study plots included turnips, beets, carrots, pattypan squash, tomatoes, garlic, strawberries, eggplant, and other brassicas. All farms reported rotating crop locations each year. Only one farm (R3) was USDA-certified Organic.

Participating farms did not consistently reply to survey questions about the scale of their operations, but one rural farm (R2) reported growing 600 square feet (0.01 acres) of kale. Another rural farm (R1) reported harvesting 150 kale plants in the previous year, less than the 600 plants harvested by one of the urban farms (U2).

Prior to planting, all farms reported applying soil amendments at least once per season, including compost, feather meal, worm castings, kelp, fish emulsion, and/or minerals. During

**Table 1. Number of kale samples by farm.** A third urban farm encountered significant pest issues that inhibited kale production; no kale was harvested from this farm and their survey results were excluded from the study. Farms were anonymized by assigning a unique identifier beginning with "U" for urban farms and "R" for rural farms.

| Farm ID | Number of field samples | Number of market samples |
|---------|-------------------------|--------------------------|
| U1 | 4 | 1 |
| U2 | 3 | 0 |
| R1 | 4 | 2 |
| R2 | 4 | 2 |
| R3 | 4 | 2 |
| R4 | 4 | 2 |

kale cultivation, both urban farms (U1, U2) and one rural farm (R3) applied compost, two rural farms applied feather meal (R2, R3), and the research farm (R4) did not apply soil amendments.

During kale growth, one urban (U2) and three rural farms (R2, R3, R4) reported using pesticides. The other urban farm (U1) relied exclusively on guardian plants (e.g., marigolds), row cover, and other non-chemical deterrents. The other rural farm (R1) used pesticides but only prior to planting. The five farms using pesticides either before or after planting used kaolin clay, neem oil, pyrethrin, spinosad, and/or *Bacillus thuringiensis*, all of which are generally allowed for use under Organic Materials Review Institute standards [58] and consequently, USDA Organic standards.

During kale cultivation, all urban farms used municipal water to irrigate. Three rural farms (R1, R2, R3) used well water and the research farm (R4) used filtered pond water. None of the urban farms and all of the rural farms used drip irrigation. Three rural farms irrigated once per week; the other farms irrigated as frequently as once per day, with variations based on weather patterns or stage of plant growth.

All farms except for the research farm (R4) identified as commercial, i.e., they grew and sold produce for profit, at least in part.

## Time from kale harvest to processing, use of refrigeration

The duration of time between harvesting kale and the point at which flash-frozen homogenized samples were put into the freezer ranged from 1.8 to 6.8 hours for field samples and 2.9 to 26.4 hours for market samples. Three of the five farms with market samples (U1, R1, R3) brought kale to the respective distribution points the day after it was harvested. Of these, two rural farms noted that they kept the kale in refrigerated storage. The urban farm did not provide any information about their storage method (the survey requested information about refrigerated transport but not storage; when designing the survey we had assumed kale would be transported directly to market following harvest), but the kale was noticeably wilted upon collection by the research team. Only one farm reported using refrigerated transport.

## Analytical quality control

Results for the quality control materials are summarized in S1 Table in S1 Data. All values were within the expected range for in-house control materials and the certified range for reference materials, with acceptable HorRat ($\leq$3.0) [59], with most <1.0.

Among the carotenoids analyzed, lutein, zeaxanthin, and total beta-carotene are reported in this study. Beta-cryptoxanthin, lycopene, and alpha-carotene were detected at trace levels (less than ~0.1 mg/100g fresh weight), but the data had insufficient precision for quantitative results. Alpha-cryptoxanthin and gamma-carotene were also monitored but not detected (<0.01 mg/100g fresh weight) in any samples (lycopene was also <0.01 mg/100g in most samples). For vitamin $K_1$, menoquinone-4 and dihydrophylloquinone were not detected (<0.1 $\mu$g/ 100 g) in any samples and thus are not reported in this study.

## Differences between urban and rural field samples

Unless specified otherwise, all analyte concentrations and statistical tests reported in the manuscript and tables refer to fresh weight concentrations. Dry weight concentrations are provided in Supplementary Figures and Tables. Mean analyte concentrations for field samples, by farm urbanicity, are reported in Table 2. Results for individual farms are reported in Figs 1–3. Additional descriptive statistics and analyte concentrations for individual samples are provided in S2-S4 Tables in S1 Data.

**Table 2. Mean fresh weight analyte concentrations in field samples by urban/urban farms, with comparisons to prior estimates.** Mean fresh weight analyte concentrations by urban/rural farm, with standard error (SE). Since there was only one market sample from urban farms, market samples are not included in means. Estimates from the current study are compared against prior estimates from the Safe Urban Harvests (SUH) study (27) and the USDA FoodData Central database entry for raw kale (NDB number 11233) (2). The SUH study included kale samples from urban farms and community gardens in Baltimore City (N = 25) and "non-urban" results that included conventional and organic kale samples from grocery stores, and samples from farmers market vendors from outside the city (N = 32). Lutein and zeaxanthin concentrations were combined for this table so they could be compared to USDA values. Chromium was not speciated between hexavalent chromium (a carcinogen) and trivalent chromium (an essential nutrient).

| Group | Analyte | Unit | Current study urban, mean ± SE | Current study rural, mean ± SE | SUH urban, mean | SUH non-urban, mean | USDA, mean (range) |
|---|---|---|---|---|---|---|---|
| | Moisture | g/100g fresh wt | 88.2 ± 0.7 | 85.2 ± 0.4 | | | |
| Carotenoids & vitamins | Beta-carotene | mg/100g fresh wt | 3.20 ± 0.26 | 5.28 ± 0.21 | | | 2.87 (2.16–3.83) |
| | Lutein + zea. | mg/100g fresh wt | 8.09 ± 0.18 | 10.65 ± 0.32 | | | 6.26 (4.46–8.56) |
| | Phylloquinone | mcg/100g fresh wt | 228 ± 5 | 350 ± 7 | | | 390 (369–422) |
| | Vitamin C | mg/100g fresh wt | 113 ± 3 | 147 ± 4 | | | 93 (84–104) |
| Nutritional elements | Calcium | mg/100g fresh wt | 397 ± 54 | 374 ± 11 | | | 254 (203–281) |
| | Copper | mg/100g fresh wt | 0.07 ± 0.00 | 0.05 ± 0.00 | 0.06 | 0.27 | 0.05 (0.03–0.08) |
| | Iron | mg/100g fresh wt | 7.58 ± 2.02 | 1.61 ± 0.06 | | | 1.60 (0.77–3.61) |
| | Magnesium | mg/100g fresh wt | 50.3 ± 6.0 | 45.3 ± 1.2 | | | 32.7 (28.4–45.8) |
| | Manganese | mg/100g fresh wt | 0.57 ± 0.07 | 0.88 ± 0.06 | 0.38 | 0.62 | 0.92 (0.51–1.46) |
| | Molybdenum | mg/100g fresh wt | 0.03 ± 0.00 | 0.02 ± 0.00 | | | |
| | Phosphorous | mg/100g fresh wt | 53.1 ± 2.1 | 52.8 ± 3.0 | | | 55 (47–62) |
| | Potassium | mg/100g fresh wt | 312 ± 15 | 441 ± 8 | | | 348 (301–389) |
| | Sodium | mg/100g fresh wt | 18.1 ± 3.3 | 8.6 ± 0.7 | | | 53 (16–107) |
| | Zinc | mg/100g fresh wt | 0.61 ± 0.02 | 0.39 ± 0.01 | 0.43 | 0.39 | 0.39 (0.20–0.57) |
| Non-essential metals | Arsenic | ppm fresh wt | 0.02 ± 0.00 | 0.01 ± 0.00 | 0.02 | 0.01 | |
| | Barium | ppm fresh wt | 3.82 ± 0.67 | 4.49 ± 0.33 | 3.37 | 2.08 | |
| | Cadmium | ppm fresh wt | 0.03 ± 0.01 | 0.02 ± 0.00 | 0.05 | 0.06 | |
| | Chromium (total) | ppm fresh wt | 0.17 ± 0.04 | 0.04 ± 0.00 | 0.05 | 0.05 | |
| | Lead | ppm fresh wt | 0.13 ± 0.02 | 0.01 ± 0.00 | 0.04 | 0.02 | |
| | Nickel | ppm fresh wt | 0.09 ± 0.02 | 0.08 ± 0.01 | 0.09 | 0.15 | |
| | Uranium | ppb fresh wt | 7.61 ± 1.52 | 5.00 ± 0.00 | | | |
| | Vanadium | ppm fresh wt | 0.12 ± 0.04 | 0.02 ± 0.00 | | | |

Since there was only one market sample from urban farms (Table 1), market samples were excluded from statistical comparisons between urban and rural kale, i.e., comparisons were only made using field samples. Since we had to use mean values for Mann-Whitney U tests, sample sizes (N = 6 mean values for each analyte, one per farm) were too small to produce statistically significant results from urban-rural comparisons, P values are provided in Figs 1–3

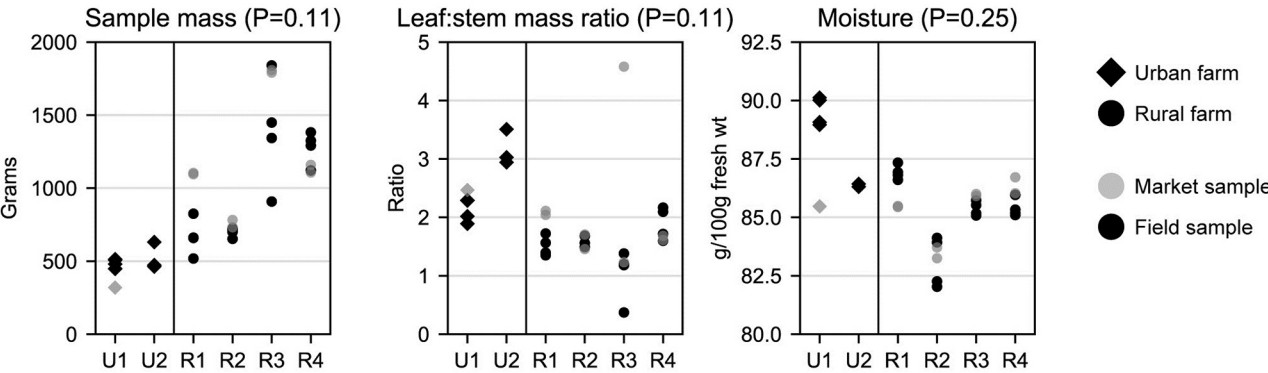

**Fig 1. Sample mass and moisture content by farm, farm type (urban vs. rural), and sampling location (field vs. market).** Each dot represents one kale sample. Site identifiers (x-axis) with "U" represent urban farms, identifiers with "R" represent rural farms. P values are from Mann-Whitney U tests comparing field samples from urban and rural farms, using the mean value from each farm (N = 6).

and S2, S3 Figs in S1 File and may be indicative of potentially meaningful differences that could be explored in larger studies.

**Stem and leaf weights.**   Sample weights (measured after removing the central stem and roots, washing, and drying) were generally lower in urban field samples, while the mass ratios of leaves to stems were generally higher in urban samples. Mean weights of rural field samples were nearly twice those of urban field samples; by contrast, the mean ratio of leaf mass to stem mass was 76% higher in urban samples.

**Moisture.**   Moisture content was similar between urban and rural samples (means: 88 and 85%, respectively). Controlling for moisture content (i.e., comparing dry weight concentrations) did not meaningfully affect the overall conclusions from any statistical analyses, including differences in analyte concentrations between urban and rural samples, differences by growing practices, and correlation tests. Time between harvesting and storing processed samples in the freezer was not correlated with moisture content (P>0.05).

**Nutrients and non-essential metals.**   Sample sizes (N = 6 mean values for each analyte, one per farm) precluded the possibility of statistically significant differences in analyte concentrations between urban and rural field samples. However, P values were as low as 0.06 for arsenic and 0.11 for all carotenoids and vitamins, copper, iron, potassium, zinc, lead, and vanadium (Figs 2 and 3). Mean concentrations of carotenoids and vitamins were generally higher in rural samples. Compared to urban field samples, mean fresh weight concentrations of carotenoids and vitamins were 22–38% higher in rural field samples. By contrast, with the exception of manganese and potassium, mean concentrations of nutritional elements were higher in urban field samples by as much as 413% for iron.

Compared to rural field samples, with the exception of barium, mean concentrations of non-essential metals were higher in urban field samples by as much as 13 times for lead and six times for vanadium (Table 2, Fig 3). That said, lead and cadmium concentrations for all samples, including urban (mean Pb: 0.13 ppm, mead Cd: 0.03 ppm) and rural (mean Pb: 0.01 ppm, mean Cd: 0.02 ppm) field samples, were below the maximum levels (Pb: 0.3 ppm, Cd: 0.2 ppm; Fig 3) specified by the Food and Agriculture Organization of the United Nations (FAO) and the World Health Organization (WHO) [60].

**Between urban-rural variance compared to within urban-rural variance.**   One-way ANOVA was used to compare variance between urban and rural farms to variance within urban and rural farms (S5 Table in S1 Data). For six elements (Ba, Ca, Cd, Mg, Ni, P), variance

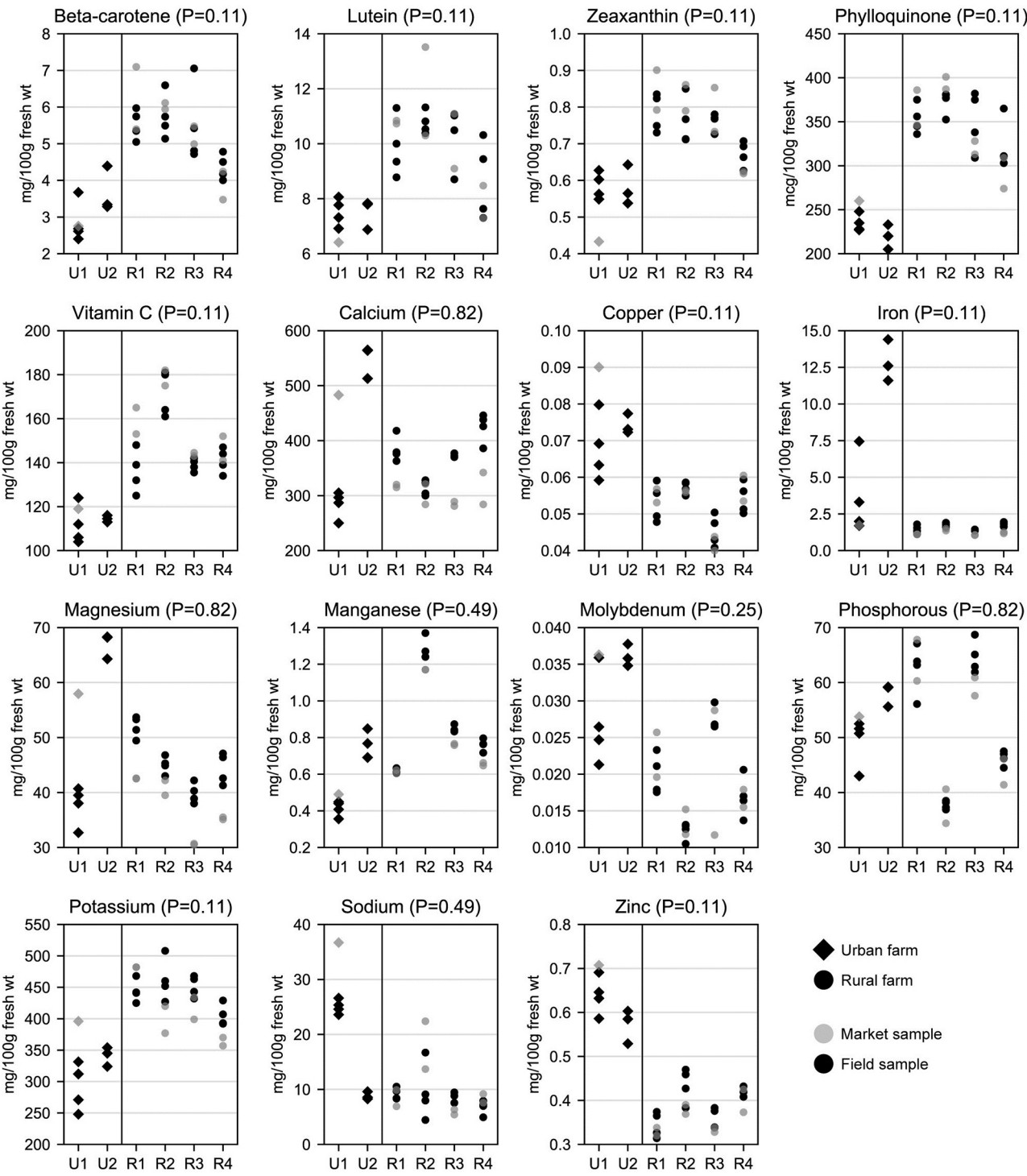

**Fig 2. Fresh weight concentrations of carotenoids, vitamins, and nutritional elements by farm, farm type (urban vs. rural), and sampling location (field vs. market).** Each dot represents one kale sample. Site identifiers (x-axis) with "U" represent urban farms, identifiers with "R" represent rural farms. P values are from Mann-Whitney U tests comparing field samples from urban and rural farms, using the mean value from each farm (N = 6). See S2 Fig in S1 File for dry weight concentrations.

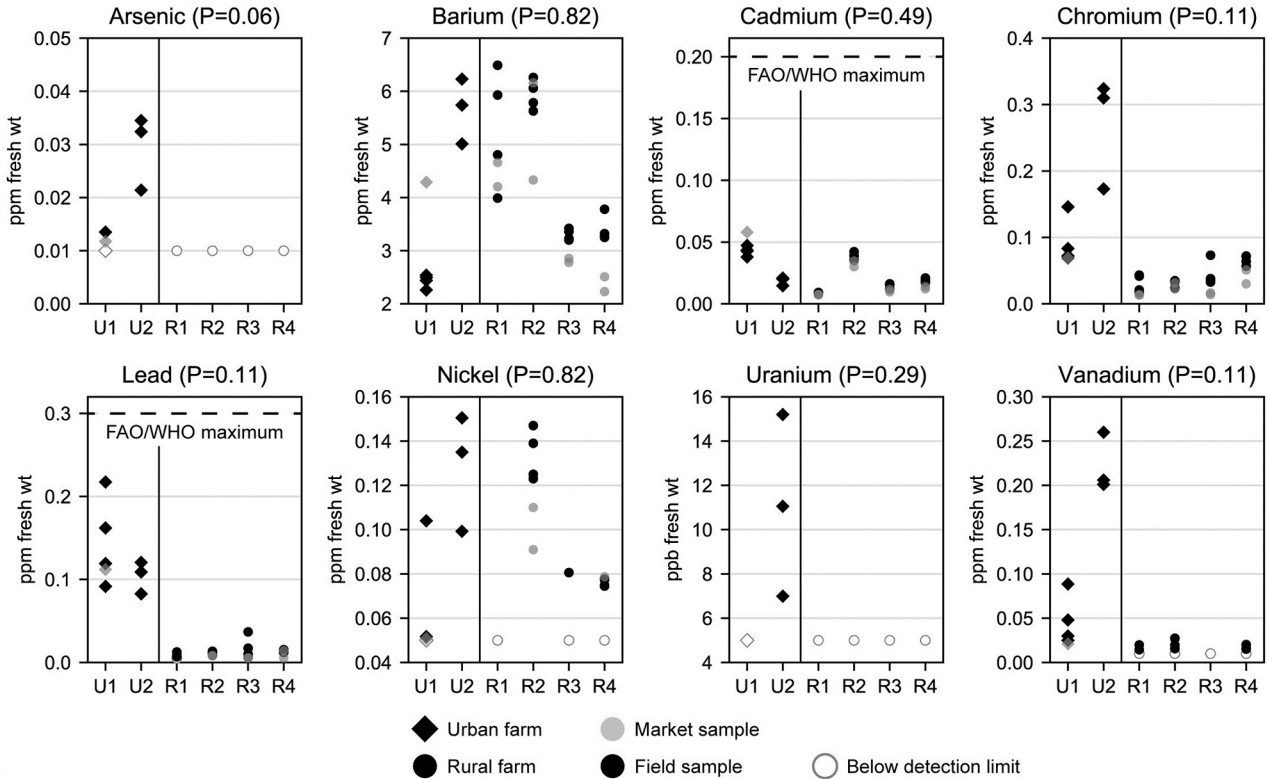

**Fig 3. Fresh weight concentrations of non-essential metals by farm, farm type (urban vs. rural), and sampling location (field vs. market).** Each dot represents one kale sample. Site identifiers (x-axis) with "U" represent urban farms, identifiers with "R" represent rural farms. P values are from Mann-Whitney U tests comparing field samples from urban and rural farms, using the mean value from each farm (N = 6). See S3 Fig in S1 File for dry weight concentrations. Chromium was not speciated between hexavalent chromium (a carcinogen) and trivalent chromium (an essential nutrient).

was greater within urban and rural farms; this can also be observed in Figs 2 and 3. These results suggest that differences in production conditions among farms may in some cases have a greater influence on analyte concentrations than farm location being rural or urban. For all 17 other analytes, however, variance was greater between urban and rural farms.

## Differences between field and market samples

The only market sample provided by urban farms was collected at the market 19 hours after harvest, and had much higher concentrations of six elements (Ca, Cu, Mg, K, Na, Ba), and lower moisture content, compared to field samples from the same farm (Figs 1–3). Given this sample was an outlier in many regards, and because there was only one market sample from urban farms (Table 1), urban farms were excluded from statistical comparisons between field and market samples.

There were no significant differences (P>0.05) in sample weight, leaf:stem ratio, or moisture content between the four mean field and four mean market samples (N = 8), by farm, among rural farms. There were also no significant differences in analyte concentrations between mean field and mean market samples (N = 8) among rural farms, with the exception of dry weight (and not fresh weight) concentrations of calcium (P<0.05). Compared to field samples, mean fresh weight concentrations of nutritional elements and non-essential metals by farm were higher in market samples by as much as 80% for lead and chromium.

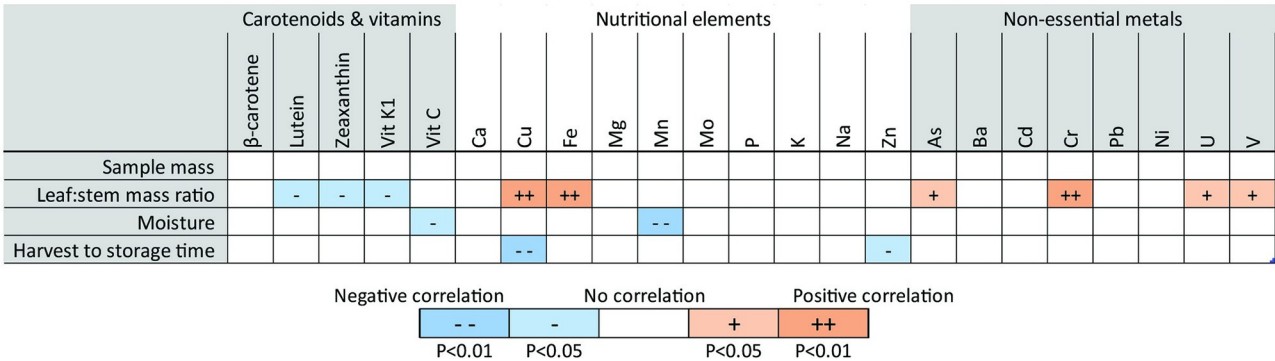

**Fig 4. Correlations between sample properties and fresh weight analyte concentrations among mean field samples by farm.** Levels of statistical significance for Pearson's correlations between sample properties (mass, moisture, and time to freezing; table rows) and fresh weight concentrations of nutrients and metals (table columns) in field samples, using the mean values from each farm (N = 6). See S4 Fig in S1 File for dry weight correlations.

Mean carotenoid and vitamins concentrations were similar between field and market samples (6% lower to 2% higher in market samples). Differences in moisture content were less than 1%.

## Correlations between mass, moisture, harvest to storage time, and analyte concentrations among field samples

The leaf:stem mass ratio was significantly positively correlated with fresh weight concentrations of some carotenoids and vitamin $K_1$, and negatively correlated with some elements and metals (Fig 4). Moisture content was significantly inversely correlated with fresh weight vitamin C and Manganese concentrations. Sample weight was not correlated with any analyte concentrations. Time between harvest and freezing samples was not correlated with concentrations of any analytes susceptible to oxidation or degradation, i.e., vitamins and carotenoids.

There appeared to be some patterns in how certain groups of analytes correlated with others (Fig 5). Fresh weight concentrations of one carotenoid or vitamin, for example, consistently tracked with the others ($P < 0.05$, top left corner of Fig 5); for example, samples high in vitamin $K_1$ were also high in carotenoids. Similarly, concentrations of nutritional elements and non-essential metals in some cases positively correlated with one another, e.g., iron with copper, magnesium, arsenic, chromium, uranium, and vanadium ($P < 0.05$). Potassium was positively correlated with carotenoids and vitamins, while lead and zinc were inversely correlated with most carotenoids and vitamins ($P < 0.05$).

## Differences in field samples by growing practice

Analyte concentrations in some cases differed significantly based on growing practices used (Fig 6). Carotenoid and vitamin concentrations, for example, were significantly different between field samples from farms that used drip irrigation vs. samples from farms that did not ($P < 0.05$), with mean concentrations higher among the former. Only rural farms reported using drip irrigation, however, thus any associations between growing practices and analyte concentrations may be confounded by urbanicity. Farms that tested their soil for metals prior to planting generally had lower concentrations of carotenoids and vitamin $K_1$, although without an obvious explanation for why this would be the case, this too is likely the result of confounding factors. No other patterns were evident.

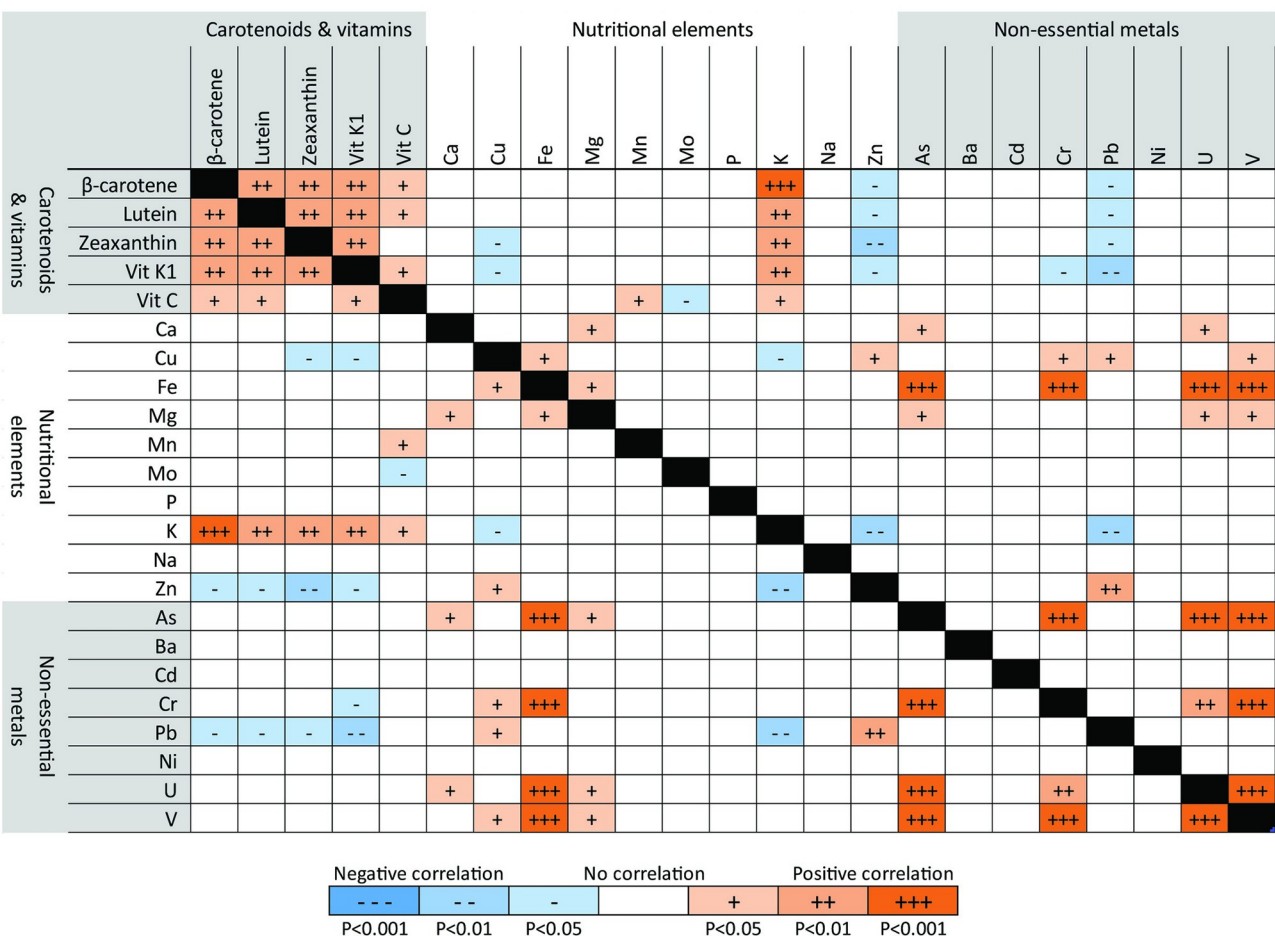

**Fig 5. Correlations between fresh weight analyte concentrations among mean field samples by farm.** Levels of statistical significance for Pearson's correlations between mean fresh weight concentrations of nutrients and metals in field samples, using the mean values from each farm (N = 6). See S5 Fig in S1 File for dry weight correlations.

## Environmental conditions

Five-minute mean particulate matter concentrations (PM$_1$, PM$_{2.5}$, respirable PM, PM$_{10}$, and total PM) are shown in in S7 Fig in S1 File. PM$_{2.5}$ concentrations ranged from 0.006–0.065 ppm (mean: 0.016 ppm); concentrations during 14 different farm visits (out of 36) were at or above EPA National Ambient Air Quality Standards (NAAQS) secondary standard of 0.015 ppm [61]. Only one farm (U2) remained below the PM$_{2.5}$ standard for all six visits. PM$_{10}$ concentrations ranged from 0.008–0.132 ppm (mean: 0.026 ppm) and were below the NAAQS secondary standard of 0.15 ppm [61]. Secondary standards are designed to protect public welfare, including protection against damage to crops. There are currently no NAAQS standards for other PM sizes. There were no significant differences in concentrations between urban and rural farms for any of the particle sizes.

Collected rainfall ranged from 0–2.2 cm per week (mean: 0.5 cm; median: 0 cm; S8 Fig in S1 File). There were no significant differences among farms or between urban and rural farms.

Peak hourly light intensity for each day ranged from 35–7739 lumens/ft$^2$ (S8 Fig in S1 File). Compared to urban farms, rural farms generally had higher light intensity readings before the

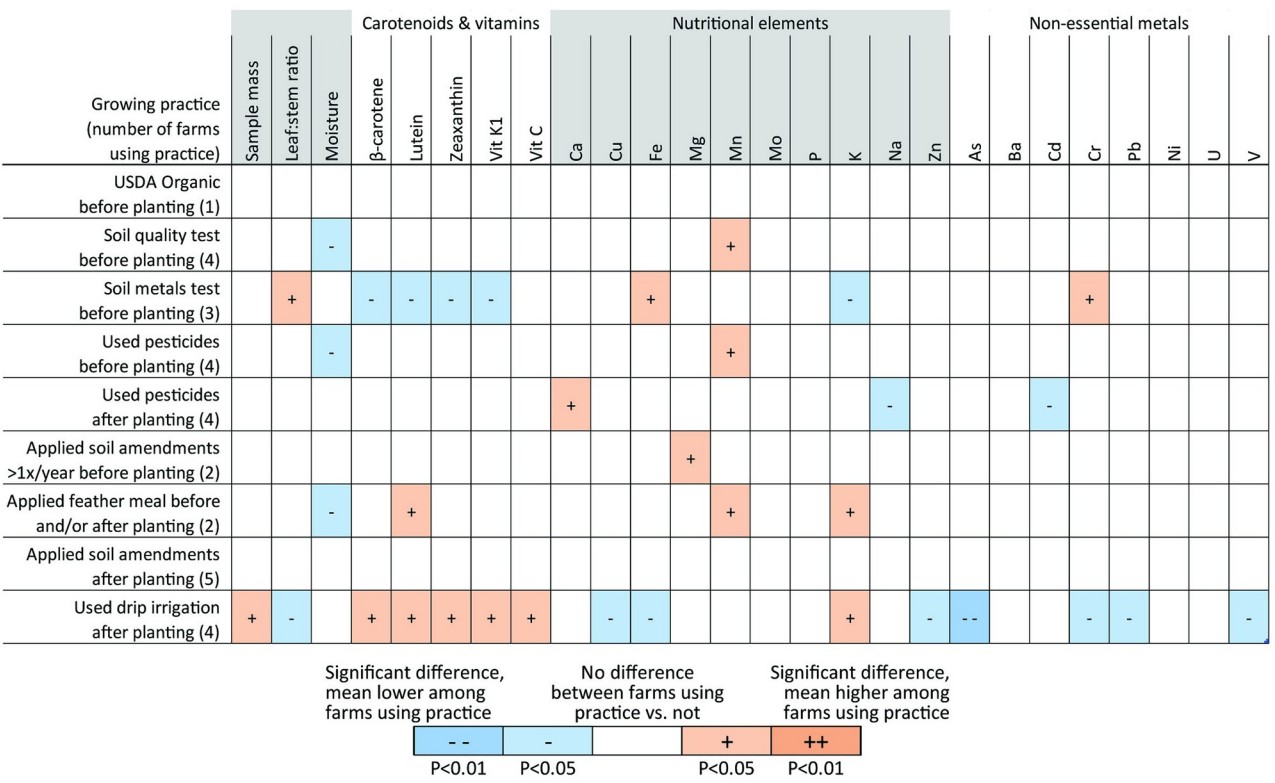

**Fig 6. Differences in mass, moisture, and fresh weight analyte concentrations by growing practice among mean field samples by farm.** Levels of statistical significance for Mann-Whitney U tests (N = 6) comparing mean mass, moisture, and fresh weight concentrations of nutrients and metals in field samples (table columns) from farms that used a growing practice vs. those that did not (table rows). Parenthesized values indicate the number of farms, out of six, following each growing practice. See S6 Fig in S1 File for dry weight differences.

third week after planting, and lower light intensity readings after the third week, continuing to decline thereafter. Overall, light intensity was significantly higher at urban farms compared to rural farms (difference in overall means: 138 lumens/ft$^2$, P<0.001), although this was likely a result of light sensors at two rural farms being in the shadow of kale leaves.

Peak hourly ambient and in-ground temperatures for each day ranged from 60–142 degrees F (16–61 degrees C) and 59–114 degrees F (15–46 degrees C), respectively, and generally declined with the change in season (S8 Fig in S1 File). Differences between urban and rural farms were highly significant (P<0.001) but negligible in magnitude (difference in overall means: <1 degree F).

## Discussion

### Comparisons to prior estimates

Since there was only one market sample from urban farms, all comparisons to prior estimates (Table 2) are based on field samples only. The Safe Urban Harvests (SUH) study analyzed concentrations of nine metals in kale (among other produce items) from urban farms and community gardens in Baltimore City (N = 25) [35]. The two urban farms in this study were also part of SUH. Although mean urban cadmium concentrations were 40 percent lower in the current study, barium, lead, and chromium were 13%, 225%, and 240% higher, respectively,

compared to SUH. This is not necessarily cause for concern given: 1) concentrations of lead in this study were below FAO/WHO standards [60] (Fig 3; cadmium levels were well below the standard, and there are no FAO/WHO standards for other metals at the time of writing); 2) levels of chromium were not speciated between hexavalent chromium (a carcinogen) and tri-valent chromium (an essential nutrient), thus if the levels in kale were predominantly com-prised of the latter there would be little cause for concern; and 3) this study only assessed kale from two urban farms, a small sample size that should not be used to make generalized conclu-sions, whereas SUH assessed kale from 25 urban farms and community gardens. Mean con-centrations of the three nutritional elements assessed in SUH (copper, zinc, and manganese) were also higher (17%, 42%, and 50%, respectively) among urban farms in the current study compared to SUH.

A more direct comparison between kale from the two urban farms in this study and kale from the same two urban farms in SUH indicates metals concentrations were, with some exceptions, higher in the current study by as much as 96% for lead (farm U1); chromium was also 82% higher in this study for one of the farms (U2). These differences may be due to a wide range of factors, including changes in soil metals concentrations (and accordingly, plant uptake) between the two study periods, either due to environmental factors and/or changes in soil management practices; spatial differences in soil metals concentrations; and/or differences in plant uptake between different kale cultivars.

The SUH study also analyzed metals concentrations in conventional and organic kale sam-ples from grocery stores, and samples from farmers market vendors from outside Baltimore City (N = 32). For five of the nine metals included in SUH, mean metals concentrations from rural farms in this study were between 20% and 81% lower compared to non-urban samples from SUH. Some differences are to be expected, in part because grocery store samples included in SUH likely represent industrial-scale farms that operate under qualitatively different condi-tions than those in the current study. Participating farms did not consistently reply to survey questions about the scale of their operations, but one rural farm (R1) reported harvesting 150 kale plants in a prior year compared to 600 plants from an urban farm (U2), while another rural farm (R2) reported growing 600 square feet (0.01 acres) of kale. For context, the average 2021 farm size in Maryland was 161 acres [62]. These data, taken together with visual observa-tions by the study team, suggest the rural farms in this study were orders of magnitude smaller than the highly mechanized, industrial-scale farms that produce the bulk of the country's vege-table output, whether organic or conventional.

Field samples from the current study were also compared to estimates of 11 nutritional components for raw kale provided in the USDA FoodData Central Foundation Foods database [2] (Table 2). Mean concentrations of beta-carotene, combined lutein and zeaxanthin, vitamin C, calcium, iron, and magnesium from both urban and rural samples in this study were higher than means reported by the USDA, and were in some cases even higher than the maximum values reported by the USDA. Iron concentrations were 374% higher in urban samples from this study compared to mean USDA values. Similar to the SUH non-urban samples, USDA estimates were based on samples from five different supermarkets across the U.S., and thus likely represent farms operating under vastly different conditions than those in the current study. Furthermore, time in transit and on supermarket shelves may have reduced the nutrient density of USDA samples. As with metals, nutrients in kale likely also vary based on variety and cultivar; this study assessed the Darkibor variety, whereas neither SUH participants nor the USDA database specify which variety was grown. Taken together with the SUH compari-sons of nutritional elements above, USDA comparisons suggest kale in some cases has the potential to be even more nutrient-dense than previously estimated, particularly if consumed shortly after harvest, regardless of urban or rural origins.

## Differences between urban and urban samples

Although small sample size (N = 6 mean values for each analyte, one per farm) precluded the possibility of statistically significant differences between urban and rural field samples, concentrations of carotenoids and vitamins were generally higher in rural samples, while concentrations of nutritional elements and non-essential metals were generally higher in urban samples, particularly for lead and vanadium. Given element concentrations in kale from urban farms in this study were generally higher than those from prior studies, and concentrations in kale from rural farms in this study were generally lower than those from prior studies, the urban-rural differences observed in this study are to be expected. For some elements, however, ANOVA results suggest differences in production conditions within a farm or among farms may have a greater influence on some analytes than farm location being rural or urban.

Neither environmental conditions nor surveys of growing practices revealed any obvious explanation as to why these differences occurred. Urban farms' use of municipal water is unlikely to explain differences in kale metal concentrations, since the overwhelming majority of irrigation water samples from the SUH study were well below public health guidelines for nine metals [35]. Since the rural farms in this study reported using drip irrigation and urban farms did not, moisture stress could partly explain the lower carotenoid, vitamin, and potassium concentrations in urban kale (Fig 6), which for potassium would be consistent with a prior study suggesting concentrations of some minerals in kale decrease as a result of moisture stress [19]. Contrary to the prior study, however, in this study concentrations of copper and iron were *higher* among kale samples experiencing greater water stress (i.e., urban kale, on the hypothesis that use of drip irrigation alleviates water stress). Further research on kale could explore the effects of these and other factors or urban-rural differences, including by analyzing irrigation water samples as in Lupolt et al. [35].

Soil conditions could also explain urban-rural differences in kale. Urban soils may be subject to numerous factors that can decrease fertility, including compaction by buildings and heavy machinery which, in turn, can decrease porosity, suppress the activity of beneficial microbes, and inhibit the accumulation of soil organic carbon [63], all of which may affect the nutrient density of urban crops. Urban areas are also often associated with sources of soil contamination [27–31] which may be taken up by plant tissues or deposited on surfaces (see Introduction). We would not expect soil on the surfaces of kale plants to be a major factor in this study given samples were thoroughly washed, although soil resides may have remained in crevices of the leaves. Follow-up studies should account for these contamination pathways by analyzing soil samples, as in Lupolt et al. [35].

To mitigate soil quality and contamination concerns, farmers may grow crops in raised beds filled with imported growing media. Both urban farms in the current study reported using some compost, but it was unclear whether they used imported growing media exclusively, or if it was added to native soil. The overwhelming majority (95%, among the 100 farms that responded to questions about soil) of Baltimore's farms and gardens participating in SUH reported growing crops in at least some soil, compost, or mulch brought in from off-site, and two-thirds (69%) used raised beds [10]. Media used to fill raised beds should be of good quality and not contaminated, although even the use of clean soils does not address the potential for soil recontamination via airborne deposition [31]. The current study did not analyze soil samples, but the SUH study in most cases did not find a significant relationship between metals concentrations in soils and those in produce; significant linear relationships were only found between soil and plant concentrations for two metals (Cu and Mn). Furthermore, all calculated bioconcentration factors (metal concentrations in produce divided by the concentration in a soil sample collected immediately adjacent to the plant [64]) from the SUH study were less

than 0.3 [35], suggesting soil element concentrations alone may be a poor predictor of concentrations in plants.

The health implications of urban-rural differences in non-essential metals are difficult to gauge without a risk assessment that would additionally consider how much kale people typically consume, among other factors. The SUH assessment of nine metals in soil, produce, and irrigation water in 104 urban farms and gardens in Baltimore City did not identify cause for concern [35], although metals concentrations in kale in this study were higher than those reported in SUH in many cases (Table 2). Any potential health implications associated with exposure to non-essential metals should not be considered in isolation, but rather balanced against the health benefits associated with diets higher in kale [1] and other vegetables.

### Differences between field and market samples

Since there was only one market sample from urban farms, statistical comparisons between field and market samples were performed on rural samples only to mitigate confounding. Since field samples were transported directly to the lab for processing and freezing, kale samples harvested from the market might be expected to have lower vitamin and carotenoid concentrations due to the oxidation and/or degradation of those nutrients. There were not, however, any significant differences in levels of potentially labile nutrients (i.e., vitamins and carotenoids) between field and market samples, nor was time from harvesting to freezing correlated with concentrations of any vitamins or carotenoids.

It is also notable that the one urban market sample had much higher concentrations of six elements and lower moisture content compared to field samples from the same farm, which would be consistent with the samples losing moisture over the 19-hour period between harvest and collection at the market, concentrating the elements.

### Correlations among analytes

We also observed some statistically significant relationships among analyte concentrations and other properties of the kale samples. The ratio of leaf mass to stem mass, for example, was negatively correlated with some carotenoids and vitamin $K_1$, and positively correlated with some nutritional elements and non-essential metals. Further research could shed light on the reasons for these relationships; studies have shown that some metals, for example, tend to accumulate at higher levels in different parts of a plant, e.g., roots vs. leaves [32–34]. This could in turn inform potential recommendations for consumers, e.g., whether there may be benefits of consuming kale stems along with the leaves.

There also appeared to be patterns in how certain groups of analytes correlated with others in field samples. Fresh weight concentrations of one carotenoid or vitamin, for example, consistently tracked with the others; e.g., samples high in vitamin $K_1$ were also high in carotenoids. Similarly, concentrations of nutritional elements and non-essential metals were in some cases correlated with one another. By contrast, concentrations of lead and zinc were negatively correlated with concentrations of carotenoids and vitamins. For lead, this is consistent with prior evidence of an inverse association between heavy metal and vitamin concentrations in plants, in part due to oxidative stress [65]. Furthermore, since urban farms were generally significantly lower in carotenoids and vitamins and higher in elements compared to rural farms, urbanicity —and the associated properties of urban and rural soils explained above—could be confounders that partly explain these correlations. Revisiting these patterns with a larger sample size could shed more light on any potential underlying phenomena that could explain them.

### Study limitations and lessons learned

The sample size for this study was small (Table 1). Larger sample sizes would be needed to control for multiple potential confounding factors at once, including urbanicity, time between harvesting and freezing samples, and growing practices. We also had only one market sample from one urban farm (which was an outlier in moisture/element concentrations) versus two market samples from each rural farm, which could skew results, thus market samples were excluded from comparisons to prior studies and from most statistical analyses. Given the popularity of farmers markets, future research could explore the potential for nutrient loss between harvest and point of sale, with an eye toward helping producers make decisions about how best to preserve nutrient density.

Some open-ended questions for growers elicited responses that were not comparable across farms because of differences in how they may have been interpreted. In response to a question about pest management, for example, some growers mentioned row covers—which are used for pest management, but also to protect against the elements and retain moisture. Differences in how growers classify the primary use of row covers could have influenced their responses, thus we did not have reliable data on the use of these and certain other practices. Wherever possible, in lieu of open-ended questions, an expanded checklist of growing and supply chain practices (e.g., use of row covers, mulching film, and refrigerated storage for market samples) for use during both questionnaires and farm visits could aid in gathering more viable data, although this would have to be balanced against potentially longer times for survey administration.

Challenges for urban farms observed during this study, including the loss of kale plants and the exclusion of one urban farm due to pest infestations, reflect the realities of growing operations that are often under resourced, understaffed, and/or run by volunteers. More extreme temperatures due to the climate crisis, combined with heat island effects in urban areas, are likely to make urban agriculture an even more challenging endeavor. Future studies on urban agriculture should plan and account for these realities, while exploring policy interventions to help address them.

### Conclusion

Kale is a hardy, resilient, and nutrient-dense crop that could serve an increasingly important role in providing an affordable, safe, and nutritious food supply, particularly within the growing urban and community farming movement, as well as in light of the climate crisis and associated effects on extreme weather patterns and food insecurity. With both urban and rural farms filling different roles and priorities in a changing food system, including climate adaptation and resilience, the safety and nutrient density of produce grown in these different environments is of critical importance. Although small sample sizes precluded statistical significance for some analyses, we observed non-significant but potentially meaningful differences between kale grown in urban and rural settings. Absent any compelling evidence to the contrary, however, the health benefits of a varied diet high in fruits and vegetables—regardless of urban or rural origins—likely outweigh any potential risks associated with exposure to non-essential metals, provided growing sites follow recommended practices for soil safety. Our findings also suggest kale in some cases may have the potential to be even more nutrient-dense than previously estimated, although those comparisons are based on a small number of fresh field samples (in this study) with retail samples (USDA) using potentially different varieties of kale. Further research with larger sample sizes could shed more light on the nutritive and metals composition of this important crop and the factors that influence it, particularly given its growing popularity.

## Supporting information

**S1 Data. S1-S5 Tables.**
(XLSX)

**S1 File. S1-S8 Figs and baseline farmer questionnaire.**
(PDF)

**S2 File.**
(DOCX)

## Acknowledgments

The authors thank the Maryland farmers who participated in the study; Johns Hopkins Center for a Livable Future's research assistants Trent Dilka, Andrea Chiger, and Ruth Young for assistance with fieldwork and other support; Nancy Pennington, who did protocol and instrument training and assisted with distributions of samples and controls to labs; Ryan McGinty, who also assisted with distribution of samples and controls, and ran the moisture and vitamin C analyses; and Kit Carson at the Johns Hopkins Institute for Clinical and Translational Research, who provided guidance on statistical methods.

## Author Contributions

**Conceptualization:** Sara N. Lupolt, Raychel E. Santo, Naomi K. Fukagawa, Katherine M. Phillips, Keeve E. Nachman.

**Data curation:** Brent F. Kim.

**Formal analysis:** Brent F. Kim.

**Funding acquisition:** Katherine M. Phillips, Keeve E. Nachman.

**Investigation:** Brent F. Kim, Sara N. Lupolt, Raychel E. Santo, Grace Bachman, Xudong Zhu, Tianbao Yang, Matthew L. Richardson, Carrie Green, Katherine M. Phillips.

**Methodology:** Brent F. Kim, Sara N. Lupolt, Raychel E. Santo, Xudong Zhu, Tianbao Yang, Carrie Green, Katherine M. Phillips.

**Project administration:** Sara N. Lupolt, Raychel E. Santo.

**Software:** Brent F. Kim.

**Supervision:** Sara N. Lupolt, Raychel E. Santo, Tianbao Yang, Keeve E. Nachman.

**Validation:** Katherine M. Phillips.

**Visualization:** Brent F. Kim.

**Writing – original draft:** Brent F. Kim, Sara N. Lupolt, Raychel E. Santo, Katherine M. Phillips.

**Writing – review & editing:** Brent F. Kim, Sara N. Lupolt, Raychel E. Santo, Grace Bachman, Xudong Zhu, Tianbao Yang, Naomi K. Fukagawa, Matthew L. Richardson, Carrie Green, Katherine M. Phillips, Keeve E. Nachman.

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
