## [Decision Letter · Decision Letter 0]

27 Oct 2023

PONE-D-23-29655Nutrients and Non-Essential Metals in Darkibor Kale Grown at Urban and Rural Farms: A Pilot StudyPLOS ONE

Dear Dr. Nachman,

Thank you for submitting your manuscript to PLOS ONE. After careful consideration, we feel that it has merit but does not fully meet PLOS ONE’s publication criteria as it currently stands. Therefore, we invite you to submit a revised version of the manuscript that addresses the points raised during the review process.

When responding to reviewer comments, please make sure to add proper/related citations to the introduction part, including also thorough discussion of the findings in terms of differences in carotenoids and vitamins vs minerals in  rural and urban settings, respectively. Your responses should also include thorough explanations of the methodology, including indicating observed levels of nutrients/elements as seen in results.

We look forward to receiving your revised manuscript.

Kind regards,

Elingarami Sauli, PhD

Academic Editor

PLOS ONE

Journal Requirements:

This work was supported by cooperative agreement 58-8040-8-018 between the US Department of Agriculture Agricultural Research Service and Virginia Tech, and by cooperative agreement 58-8040-8-021 between the U.S. Department of Agriculture Agricultural Research Service and Johns Hopkins University. Sara Lupolt was supported by a dissertation grant from the Johns Hopkins 21st Century Cities Initiative; a pilot award from the Johns Hopkins Education and Research Center for Occupational Safety and Health, supported by the National Institute for Occupational Safety and Health and the US Department of Agriculture Northeast Sustainable Agriculture Research and Education Program (GNE 19-209); and a Johns Hopkins Center for a Livable Future-Lerner Fellowship. Because this was a cooperative agreement, three USDA staff members (Naomi Fukagawa, Carrie Green, and Tiabao Yang) were involved in helping design, conduct, and co-author the study. Other funders had no role in preparing, reviewing, or editing the manuscript. Support for statistical consulting was made possible by The Johns Hopkins Institute for Clinical and Translational Research, funded in part by [Grant Number UL1 TR003098] the National Center for Advancing Translational Sciences, a component of the National Institutes of Health (NIH) and the NIH Roadmap for Medical Research.

Reviewers' comments:

Reviewer's Responses to Questions

**Comments to the Author**

1. Is the manuscript technically sound, and do the data support the conclusions?

Reviewer #1: Yes

Reviewer #2: Yes

Reviewer #3: Yes

Reviewer #4: No

Reviewer #5: Yes

Reviewer #6: Yes

2. Has the statistical analysis been performed appropriately and rigorously? 

Reviewer #1: Yes

Reviewer #2: Yes

Reviewer #3: Yes

Reviewer #4: No

Reviewer #5: No

Reviewer #6: Yes

3. Have the authors made all data underlying the findings in their manuscript fully available?

Reviewer #1: No

Reviewer #2: Yes

Reviewer #3: Yes

Reviewer #4: Yes

Reviewer #5: Yes

Reviewer #6: Yes

4. Is the manuscript presented in an intelligible fashion and written in standard English?

Reviewer #1: No

Reviewer #2: Yes

Reviewer #3: Yes

Reviewer #4: Yes

Reviewer #5: Yes

Reviewer #6: Yes

5. Review Comments to the Author

Reviewer #1: General comment:

The pilot study discussed in this paper examines the nutritional elements, non-essential metals, and health-relevant nutrients in Darkibor kale grown in urban and rural settings. Despite its small sample size, the research reveals significant differences in kale composition between these two environments, which has important implications for the future of kale farming and nutrition.

Specific comments:

1. The paper effectively underscores the significance of kale as a nutrient-dense vegetable and its potential role in addressing the challenges posed by the climate crisis. This context is essential for understanding the importance of the study. What are the specific nutritional elements and non-essential metals that were found to exhibit differences between urban and rural-grown Darkibor kale in the study?

2. The study's limitations in terms of sample size are acknowledged, but it does a commendable job in highlighting potentially meaningful differences in kale composition between urban and rural farms. However, more details about the sample size and methodology would be beneficial for readers to gauge the study's reliability.

3. It's noteworthy that the research identifies higher concentrations of elements in kale from urban farms and elevated levels of carotenoids and vitamins in kale from rural farms. This specific insight adds value to our understanding of kale quality and can guide both urban and rural farming practices. The mention of variance within and between farm settings is an interesting observation that could lead to further research. Can you elaborate on the potential reasons behind the variations in kale composition between urban and rural farming environments, and how might these findings impact farming practices?

4. This paper sheds light on the nutritional composition of Darkibor kale grown in urban and rural settings, highlighting distinctions between the two. Despite the small sample size, the study presents valuable insights into kale quality and the factors contributing to these differences. Considering the importance of kale in addressing the climate crisis and food security, what future research or actions do you believe are necessary based on the insights gained from this pilot study?

Constructive feedback:

The paper could be strengthened by providing a more comprehensive explanation of the methodology employed in the study. Additionally, including data on the actual concentrations of the identified elements and nutrients would help readers better grasp the significance of the findings.

While the study rightly mentions that concentrations of metals remained below public health guidelines, it would be beneficial to explicitly state the actual concentrations and compare them to those guidelines for clarity.

Summary:

The significance of kale as a climate-resilient crop is well articulated, and the study motivates further investigation in this area, which is crucial given the increasing popularity of this nutrient-dense vegetable. However, to improve its impact, the paper should provide more detailed information on the methodology and specific concentration levels of elements and nutrients.

Reviewer #2: The manuscript entitled “Nutrients and Non-Essential Metals in Darkibor Kale Grown at Urban and Rural Farms: A Pilot Study” has been reviewed. The manuscript is well written; however, before the final acceptance authors should execute the following minor corrections:

1. The abstract is too much descriptive. Please add some statistical findings.

2. Line 124, “Between 27-29 August” please mention the year.

3. Line 132, “Supporting” should be “supporting”.

Reviewer #3: This study explored potential differences in nutritional elements, non-essential metals, and certain health-relevant nutrients in Darkibor kale grown at urban versus rural farms. The objectives are clear. The selection of farms for planting and survey are good representatives. The sample harvesting and processing, the analyses of different parameters are well-described. The results collected are meaningful. Although there are constraints such as small sample sizes, which precluded statistical significance for some analyses, this study identified non-significant but potentially meaningful differences in in concentrations of several components between urban and rural kale samples. The paper is well-written. It can be accepted for publication in “PLOS ONE”: after making the following revision:

1. Line 48: “According to …”.

2. Line 126: “All plants were grown in open air, …” Is it possible to provide the growth conditions of different farm?

3. Table 2: It would be better to include standard errors and statistical analysis for the means.

4. Lines 419 - 421: “…contrast, with the exception of manganese and potassium, mean concentrations of nutritional elements were higher in urban field samples by as much as 413% for iron. Why? Was the lower iron concentration of kale from the rural resulting from iron deficiency?

5. Lines 468 – 471 “Similarly, concentrations of elements and metals in some cases positively correlated with one another, e.g., iron with copper, magnesium, arsenic, chromium, uranium, and vanadium (P<0.05). Potassium was positively correlated with carotenoids and vitamins, while lead and zinc were inversely correlated with most carotenoids and vitamins (P<0.05).” Why?

6. Lines 470-471: “Potassium was positively correlated with carotenoids and vitamins, while lead and zinc were inversely correlated with most carotenoids and vitamins (P<0.05). Why? Any explanation with references to support?

7. Lines 635-639: “By contrast, concentrations of lead and zinc were negatively correlated with concentrations of carotenoids and vitamins. Since urban farms were generally significantly lower in carotenoids and vitamins and higher in elements compared to rural farms, urbanicity—and the associated properties of urban and rural soils explained above—could be the confounders that explain these differences.” From Table 2, kale grown in urban farm had much higher Zinc that rural farm. Why urban farms were generally significantly lower in carotenoids and vitamins? Were the lead and zinc concentrations in urban farm soil too high, which resulted in the reductions of carotenoids and vitamins?

Reviewer #4: PONE-D-23-29655-Nutrients and Non-Essential Metals in Darkibor Kale Grown at Urban and Rural Farms: A Pilot Study

The words “Pilot study” may not be relevant in the publications.

Abstract

Lines 32-34 “In this pilot study we explored potential differences in nutritional elements, non-essential metals, and select health-relevant nutrients in Darkibor kale grown at urban versus rural farms”

What is the meaning of “pilot study” in this context?

What do authors mean by “Darkibor kale”- I guess Darkibor is a local name.

Lines 35-37 “Although small sample sizes precluded statistical significance for some analyses, we identified non-significant but potentially meaningful differences in concentrations of several components between urban and rural kale samples”

Clarify the meaning of “small sample sizes precluded statistical significance for some analyses’

The statements like “we identified non-significant but potentially meaningful differences…” This is quite confusing. Please present direct and explicit information, and given that this is an abstract, then should use catchy statements.

Lines 38-41 “Some urban-rural differences were more pronounced than those identified in prior research, although for some elements, variance within urban and rural farms was greater than variance between urban and rural farms”

Please rephrase the statement. See the bolded section.

Line 45: Introduction

This section was fairly well written though there were no compelling reasons to warrant research in Urban and Rural set up. For instance, are there more fumes as a result of vehicles and hence more heavy metal in urban areas? Let the background show previous work be reviewed showing the critical dissimilarities and find the gap that this work intended to fill. Please also clearly show the objective(s) of study and hypothesis/hypotheses (these should come out at the end of the section).

Line 102: Materials and Methods

This section is quite mixed up and quite confusing. This was supposed to be experimental research. Whereas in rural areas there were four (4) experimental farms, there were only two experimental farms in the rural area. Why this differences? My guess would be that the farms formed part of replications.

The experimental design is not shown but even more fundamental is that the replication does not comes out. For the rural areas, we assume that the four farms represent 4 replicates; which make lots of sense. However, the same cannot be said about the urban area. The two farms cannot be considered as a sufficient replication as this is not analysable.

The authors did not take initial soil samples before the start of the experiments. In that case it is not possible to speculate the possible causes in differences in kales grown in the urban areas compared to rural areas.

Line 103 Farm selection

It is not clear why this is being reported in the manuscript. I fully acknowledge the effort made in the identification of the farms, but the information is not necessary for the purposes of this manuscript. The authors can explain how the population in the rural and urban affect the mineral elements in the soil and hence uptake by the plant varieties.

Line 114 Kale seedling and planting

This subtitle is confusing – unless there were two sets where there was direct planting and transplanting of the seedlings. Can the authors clarify this?

Line 117 There is need to clarify the statement “The seeds were sown in seedling starting trays filled with Fafard growing mix” – see the bolded words

Lines 125-126 “in configurations most suitable for the location, e.g., 1x24, 2x12, 3x8 or 4x6” – the authors need to clarify the spacing as shown in bold above. What these spacing/numbers stand for?

Line 130 Surveys on farm history and growing practices

This section may not be relevant unless during the process (survey), soil samples were picked and analysis made to check the chemical composition (including essential elements, beneficial elements, non-essential or toxic heavy metals and pesticides). Otherwise the way the section is presented, I do not find its relevance, but the authors can explain.

Line 161 “Kale sample processing and storage”

The authors need to justify the relevance of this section. Is it important that the vegetables be transported in refrigerated form yet tissue analysis (usual procedures) require heating at very high temperatures?

Line 180: Nutrient and non-essential metals analyses

In this section, there is need to clearly break down the methods- they cannot be lumped together- e.g. the methods for extraction of heavy metals is quite different compared for instance with the other essential elements. Method for P is totally different from say Ca and so they should be done as subsections. Then the vitamins and carotenoids are also determined using different protocols.

Still under this section, please explain the need for the vegetables being carried in ice box or freezer.

But looking at section starting from line 197- “analytical methods”, then you will notice that there is no need for section on “Nutrient and non-essential metals analyses” above since section starting from line 197 is more comprehensive. Maybe have section as “Determination of minerals and vitamins in vegetables” to replace “analytical methods” and do away with “Nutrient and non-essential metals analyses”

However section in lines 209-213 need more elaboration – the analysis part of the metals is clear but the extraction methods not mentioned. A brief mention of the methods with citation of authors that the methods are adapted from can be quite useful.

Line 241: Data reporting and analyses

The contents should have been better be placed under “Results” section starting from section 286.

Why report in ppb instead of ppm? No wonder the metals were below the detectable limit leading to erroneous conclusion that the vegetables are safe (irrespective of whether cultivated in the urban or rural area (see abstract; line 41).

Line 255- there is a formula for conversion from fresh weight to dry weight- why not just dry the vegetables at a prescribed temperature till the weight is stable?

Line 286 Results

Line 287 Kale yields

In this section, reference is made of field samples and market samples – from materials and method section, such categorization was not evident. Can this be elaborated?

Table 1; These are raw data that have not been analysed and hence there is no point of presenting in this section.

Line 301: Farm history and growing practices

The section does fit under results – moreover, this would only be important if that history led to differences in factors like pH, organic carbon and nutrients as these would affect the availability of these elements to the crop. Can the authors explain the relevance of this section as far as the presentation of the results are concerned

Line 338; Kale sample processing and storage

This section does not belong to the “results” section. Is there any reason as to why this section is presented under “results” section?

Line 348: Analytical quality control

The authors present quite interesting results on different types of carotenoids. Unfortunately such differentiations/distinctions were not presented under “materials and methods’ section.

Table 2: the heavy and essential elements are reported in terms of mg/100mg fresh weights. It is not common to report metal concentrations in fresh weights, instead dry weight is used. Any reasons for this exception?

And for the case when the concentration is in ppm fresh weight – why not mg/kg since we are dealing with solid particles and not liquid?

Note that here the reporting is in ppm yet in the materials and method the authors always made reference of ppb.

There is a column for “current mean- Urban” and “current mean, rural” – were these results analysed? Please use t-test to compare the two. But since the mean for urban had can only two replications, is it possible to have calculation using two replications?

In the figures, the symbols U1, U2, R1, R2, R3 and R4 are used yet there was no mention of these under “materials and method” section. It would have been polite for the authors to have figure captions where all these could be explained. Otherwise the way they are, it is not possible to interpret.

It is quite interesting that almost all the results show p-values more than 0.05 hence not significant. Is there possible reason? Could it be that calculation was not well done?

Discussion

Since the methodology needs serious reworking (a lot need to be cut out – especially the survey part and only remain with experimental components). And once the result section is more comprehensive and clear (cut out on survey and reduce the subsections under results; same with discussion). Once there is that kindly harmony, it is too premature to take a look at the discussion.

Reviewer #5: The material and method section is too narrative, and contains a lot of redundant information. It should be more concise and clear. E.g. Line 135 Redundant information. Line 162 the same.

Why authors choose different digestion methods for essential vs trace elements analysis?

Lines 516-519 already mentioned, not part of the discussion

Reviewer #6: In the manuscript entitled “Nutrients and Non-Essential Metals in Darkibor Kale Grown at Urban and Rural Farms: A Pilot Study”, the authors studied a pilot plant to explore potential differences in nutritional elements, non-essential elements, and some health-relevant nutrients in kale grown at urban versus rural farms. The manuscript is very relevant and interesting. However, the authors need to clarify some parts before publication on Plos one.

The following are my comments and suggestions:

General comments:

The authors need to revise all the units and the effective numbers in the whole manuscript.

Abstract:

The abstract needs to contain all parts of the manuscript. So the authors need to improve this topic, including parts of the methodology. The authors need to insert some results (numbers) , for example, when the authors said “Concentrations of elements were generally higher in kale from urban farms”, how much higher? A conclusion sentence about their work is necessary too.

Introduction:

The introduction is interesting and contextualizes the environmental problem for the reader. However, the authors should insert more recent references in the Introduction and it´s essential to describe the potentially toxic metal levels allowed in the national and international legislation. Also, when the authors said on page 4 lines 83-84“concentrations of essential and non essential (arsenic (As), barium (Ba), cadmium (Cd), chromium (Cr), lead (Pb)) metals in..” trivalente chromium is essential for the human being. The authors need to research more specific references about it. Please, pay attential to this part including about chemistry speciation.

Page 5: “Building upon the existing body of evidence..” The authors need to write more about the aim of the work. They determined a lot of importance parameters obtaining different kind of results, but what is the main question to be answered?

Materials and methods:

Page 06: The authors need to standardize the date. Sometimes the authors write day, month and year and other times no.

Page 10: The authors need to explain more about the digestion process, including more information as the weight of the samples, temperature, final volume, acid concentration, etc.

Page 10: The authors need to write more about organic and inorganic determination as lines of emissions, analytical concentration curve, detection and quantification limits. The authors need to mention the replicates here too.

Page 12, line 255: The authors need to include the reference for this equation.

Results and discussion

In general, the authors need to discuss more the results comparing with the literature. The authors need to explore further the parts containing discussion and results, for example, in the topic “Correlations among analytes”, the authors said “We also observed some statistically significant relationships among analyte concentrations and other properties of the kale samples.”, the authors need to discuss and to say more specifically about all the parameters studied. In the sentence, “Similarly, concentrations of elements and metals were in some cases correlated with one another. “, the authors need to specify the elements here. What´s the diference between metals and elements?

6. PLOS authors have the option to publish the peer review history of their article (what does this mean?). If published, this will include your full peer review and any attached files.

Reviewer #1: **Yes: **Wiktor Halecki

Reviewer #2: **Yes: **Tapos Kormoker

Reviewer #3: No

Reviewer #4: **Yes: **Joseph Gweyi-Onyango

Reviewer #5: No

Reviewer #6: No

---

## [Author Response · Author response to Decision Letter 0]

11 Dec 2023

Please refer to the attached PDF for responses to reviewer comments.

---

## [Decision Letter · Decision Letter 1]

20 Dec 2023

Nutrients and Non-Essential Metals in Darkibor Kale Grown at Urban and Rural Farms: A Pilot Study

PONE-D-23-29655R1

Dear Dr. Keeve. E. Nachman,

We’re pleased to inform you that your manuscript has been judged scientifically suitable for publication and will be formally accepted for publication once it meets all outstanding technical requirements.

Kind regards,

Elingarami Sauli, PhD

Academic Editor

PLOS ONE

Additional Editor Comments (optional):

Reviewers' comments:

Reviewer's Responses to Questions

**Comments to the Author**

1. If the authors have adequately addressed your comments raised in a previous round of review and you feel that this manuscript is now acceptable for publication, you may indicate that here to bypass the “Comments to the Author” section, enter your conflict of interest statement in the “Confidential to Editor” section, and submit your "Accept" recommendation.

Reviewer #3: All comments have been addressed

Reviewer #7: All comments have been addressed

2. Is the manuscript technically sound, and do the data support the conclusions?

Reviewer #3: Yes

Reviewer #7: Yes

3. Has the statistical analysis been performed appropriately and rigorously? 

Reviewer #3: Yes

Reviewer #7: Yes

4. Have the authors made all data underlying the findings in their manuscript fully available?

Reviewer #3: Yes

Reviewer #7: Yes

5. Is the manuscript presented in an intelligible fashion and written in standard English?

Reviewer #3: Yes

Reviewer #7: (No Response)

6. Review Comments to the Author

Reviewer #3: The authors have addressed all of my concerns with the original manuscript. The revised manuscript can be now accepted for publication.

Reviewer #7: Authors have addressed all the comments and suggestions of all reviewers and have done good efforts for improving the manuscript. Therefore, I consider that it is now I consider that their R1 version of the manuscript may be accepted for publication.

7. PLOS authors have the option to publish the peer review history of their article (what does this mean?). If published, this will include your full peer review and any attached files.

Reviewer #3: No

Reviewer #7: No

---

## [Editor Report · Acceptance letter]

2 Apr 2024

PONE-D-23-29655R1 

PLOS ONE

Dear Dr. Nachman, 

I'm pleased to inform you that your manuscript has been deemed suitable for publication in PLOS ONE. Congratulations! Your manuscript is now being handed over to our production team.

Kind regards, 

on behalf of

Dr. Elingarami Sauli 

Academic Editor

PLOS ONE